# Effect of Curing Temperature of Epoxy Matrix on the Electrical Response of Carbon Nanotube Yarn Monofilament Composites

Omar Rodriguez-Uicab *, Tannaz Tayyarian and Jandro L. Abot

Department of Mechanical Engineering, The Catholic University of America, Washington, DC 20064, USA; 63tayyarian@cua.edu (T.T.); abot@cua.edu (J.L.A.)
* Correspondence: rodriguezuicab@cua.edu

**Abstract:** In order to evaluate the capability of carbon nanotube yarn (CNTY)-based composites for self-sensing of temperature, the temperature-dependent electrical resistance of CNTY monofilament composites was investigated using two epoxy resins: one that cures at 130 °C (CNTY/ERHT) and one that cures at room temperature (CNTY/ERRT). The effect of the curing kinetics of these epoxy resins on the electrical response of the embedded CNTY was investigated in prior studies. It was observed that the viscosity and curing kinetics affect the level of wetting and resin infiltration, which govern the electrical response of the embedded CNTY. In this work, the cyclic thermoresistive characterization of CNTY monofilament composites was conducted under heating–cooling, incremental heating–cooling, and incremental dwell cycles in order to study the effect of the curing temperature of the epoxy matrix on the electrical response of the CNTY monofilament composites. Both monofilament composites showed nearly linear and negative temperature coefficients of resistance (TCR) of $-7.07 \times 10^{-4}$ °C$^{-1}$ for specimens cured at a high temperature and $-5.93 \times 10^{-4}$ °C$^{-1}$ for specimens cured at room temperature. The hysteresis loops upon heating–cooling cycles were slightly smaller for high-temperature cured specimens in comparison to those cured at room temperature. A combination of factors, such as resin infiltration, curing mechanisms, intrinsic thermoresistivity of CNTY, variations in tunneling and contact resistance between the nanotubes and CNT bundles, and the polymer structure, are paramount factors in the thermoresistive sensitivity of the CNTY monofilament composites.

**Keywords:** carbon nanotube yarn; electrical resistance; epoxy resin; monofilament composites

## 1. Introduction

The excellent mechanical, electrical, and thermal properties of carbon nanotubes (CNTs) are leading to an increase in their applications in composite materials with sensing and actuating capabilities [1,2], structural health monitoring [3,4] and electrical devices [5,6]. The realization of these applications requires these properties to be translated from nanometric to macroscopic materials. Carbon nanotube yarns (CNTYs) are produced by liquid- or solid-state spinning processes and are promising materials due to their high thermal [7,8] and electrical properties [9,10]. CNTYs show lower physical properties than their individual CNTs [9,11]. Potential applications of CNTYs include damage monitoring in composite materials [12,13], electronics devices [14], energy storage [15], artificial muscles [16], and strain sensing [17]. A considerably less investigated potential application for CNTYs is as temperature sensors, using their intrinsic thermoresistivity while integrated in polymeric materials [9,18]. The polymer matrix plays a key role in the mechanical stability of these sensors [19]. Epoxy resins are typically known by their intrinsic brittleness due to the high crosslinking density formed during curing [20,21]. The final properties of a structural epoxy system are not only highly influenced by the type and chemical structure of the monomers and the curing agent, but also by the curing conditions and external factors, such the curing temperature, pressure, and so forth [20,21]. The effect of the chemical structure of the monomer and the curing temperature in the crosslinking structure in epoxy

resins have been investigated in several works [22,23]. In particular, the use of low curing temperatures can yield a thermosetting resin with a low glass transition temperature ($T_g$), since some reactive groups, either from the epoxy resins or curing agents, do not react completely [24]. In contrast, the application of high-temperature curing may result in the ultimate conversion of the polymer system but may also change the final network structure by allowing it to relax and explore more conformational space between polymeric chains [25]. Lambert et al. [26] analyzed the effect of the curing temperature in the crosslinking density of epoxy resin. Their results showed high crosslinking density for specimens cured using a high curing temperature, as indicated by the increase of $T_g$ and the decrease in reaction exotherms of the epoxy cured at a medium temperature (90 °C). Gupta and Brahatheeswaran [20] proposed that increasing the curing temperature of epoxy resins generally yields increased polymer crosslinking density. Lascano et al. [21] investigated the effect of the curing temperature on the density, rheological, morphological, mechanical, and thermomechanical properties of epoxy resins. Their results showed higher density and mechanical properties in polymers cured at medium temperatures in comparison to polymers cured at lower temperatures. Crasto and Kim [22] analyzed the electrical resistance changes of carbon fiber and epoxy resin monofilament composites cured at room temperature (RT) and medium temperature. In their work, the composites cured at medium temperature showed higher electrical resistance changes in comparison to composites cured at RT. The electrical characterization of CNTYs during temperature variations has been investigated [7,8], but the understanding of their electrical response under incremental heating–cooling cycles using polymers cured at different temperatures has not been thoroughly studied. Monofilament composites were used to obtain the electrical resistance response of CNTYs using epoxy resin matrices with different curing temperatures. The intrinsic thermoresistivity of CNTYs and the resin infiltration of the CNTY and the resin crosslinking are used to explain the analyzed thermoresistive response.

## 2. Materials and Methods

### 2.1. Materials

The CNTY used in this study was fabricated from a vertically aligned carbon nanotube (CNT) array with no post-processing at Nanoworld Laboratories (University of Cincinnati, Cincinnati, OH, USA). The Si wafer, including a 5 nm alumina buffer layer and 1.2 nm catalyst, was loaded into a chemical vapor deposition (CVD) reactor. The growth of CNTs on the catalyst sites of the Si wafer was achieved by heating the reactor to 750 °C in the presence of Ar and $C_2H_4$. The CNT array detachment was achieved by delivery of Ar and $H_2O$ during cooling. The CNT array, consisting of multi-wall CNTs, was spun into a yarn. The diameter, density, angle of twist, and average electrical resistivity of the densified CNTYs were ~30 μm, ~0.65 g/cm$^3$, ~30°, and $1.7 \times 10^{-3}$ Ω cm, respectively [27]. Figure 1 shows an image of the twisted yarn obtained by scanning electron microscopy (SEM) at $5000\times$ magnification.

Two commercial thermosetting epoxy resins (ERs) with different curing mechanisms were used as polymeric matrices, viz. Epon 862 (Miller-Stephenson Chemical Co., Danbury, CT, USA) and Toolfusion (Airtech International Inc., Huntington Beach, CA, USA) [28,29]. Epon 862 (ERHT) was mixed with its curing agent (Epikure W) with the stoichiometric ratio of 100:23 by weight for 3 min. The mixture was cured at 130 °C for 1.5 h. The stoichiometric ratio of Toolfusion (ERRT) 1A/1B was 100:20 by weight and was cured at room temperature (RT ~25 °C) for 10 h. The coefficient of thermal expansion of Epon 862 and Toolfusion was $57.8 \times 10^{-6}$ K$^{-1}$ and $59.4 \times 10^{-6}$ K$^{-1}$, respectively [30].

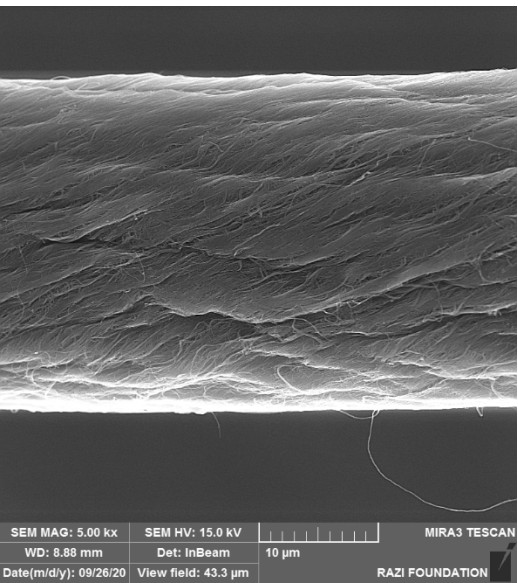

**Figure 1.** Scanning electron microscopy of CNTY at 5000×.

### 2.2. Manufacturing of CNTY/Epoxy Monofilament Composites

Four parallel 40 AWG copper wires were used for four-point probe electrical measurements and horizontally fixed into a silicon mold using a needle. Then, the CNTY was inserted longitudinally on top of the copper wires and bonded to the wires using carbon-black-based conductive paint (Bare Conductive, London, UK). The CNTY was pre-stressed with a constant mass of 116 mg (~0.1% of the CNTY tensile strength [30]) before attaching it to the mold to ensure the same condition in all the specimens. For ERHT specimens, the resin and Epikure W were mixed for 3 min and the mixture was preheated at 100 °C for 15 min to remove air bubbles. The preheated mixture was poured into the silicon mold containing the CNTY and electrodes. Then, the mold was inserted inside a Fisher Scientific Isotherm 625 G convention oven (Rockville, MD, USA) and heated for 1.5 h at 130 °C [30]. The oven was turned off after the elapsed time of curing and the specimens were cooled down to RT, with the process lasting about 5 h. For CNTY/ERRT monofilament composite specimens, resin and hardener (Toolfusion 1A/1B) were mixed for 3 min and placed in a vacuum chamber to remove the air bubbles. The mixture was poured into the silicon mold and cured for 10 h at RT. Figure 2 shows the specimen dimensions along with the experimental setup of the in situ electrical and temperature measurements during the cyclic thermoresistive characterization of CNTY monofilament composites. Three replicates of each specimen were fabricated for each test plan.

### 2.3. Experimental Setup for Electrical Resistance Characterization

The electrical resistance ($R$) of the CNTY monofilament composites was measured during the temperature programs using a NI PXI-4072 (Austin, TX, USA) impedance-capacitance-resistance (LCR) card mounted in a NI PXI-1033 chassis, as shown in Figure 2. The four-point probe measurement technique was used to calculate $R$ by measuring the voltage drop ($V$) in the inner electrodes while applying a constant current ($I$). This method was chosen to ensure that the electrical resistance of the CNTY was not affected by the contact resistance between the conductive paint electrodes and the CNTY [30]. The temperature ($T$) was measured during the experiment using a K-type thermocouple placed close to the CNTY. The thermocouple was connected to a NI 9211 card mounted in a NI cDAQ-9178 chassis. The data acquisition was conducted at 1 Hz using NI Signal Express software.

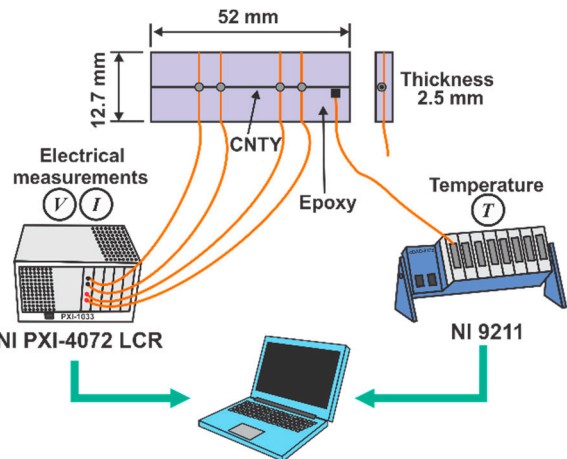

**Figure 2.** Dimensions and in situ electrical and temperature measurements of CNTY monofilament composites.

Heating–cooling and incremental heating–cooling cycles were used for thermoresistive characterization. The heating–cooling cycles were started by heating the specimens from RT ($\sim$25 °C) to 80 °C and cooling back to RT for four continuous cycles. The second program started by heating the specimens from RT to 35 °C and cooling back to RT (Cycle 1). The specimens were heated again from RT to 45 °C and cooled back to RT (Cycle 2). Finally, Cycle 3 started by heating the specimens from RT to 55 °C and cooling back to RT. For consistency purposes, the maximum temperature was considered to be 80 °C for both material systems (CNTY/ERRT and CNTY/ERHT) so that neither thermoresistive response could be affected by the thermal degradation of the polymeric matrix.

The heating temperature coefficient of resistance ($\beta_H$) was calculated at heating sections as the slope of the fractional change in electrical resistance versus the temperature change in the heating zones, i.e.,

$$\beta_H = \frac{\left(\frac{\Delta R}{R_0}\right)_{\text{Heating}}}{\Delta T} \tag{1}$$

where $(\Delta R/R_0)_{\text{Heating}} = (R_{i+1} - R_i)/R_i$ and corresponds to the fractional change in electrical resistance in ramping zones, and $\Delta T = T_i - T_0$ is the corresponding change in temperature.

The cooling temperature coefficient of resistance ($\beta_C$) was calculated at cooling zones according to:

$$\beta_C = \frac{\left(\frac{\Delta R}{R_0}\right)_{\text{Cooling}}}{\Delta T} \tag{2}$$

where $(\Delta R/R_0)_{\text{Cooling}} = (R_{i+1} - R_i)/R_i$ and corresponds to the fractional change in electrical resistance during cooling zones.

The thermoresistive hysteresis of CNTY monofilament composites was quantified using two parameters: the residual fractional change in electrical resistance $(\Delta R/R_0)_{\text{Res}}$, which is defined as the difference between the initial and final values of $\Delta R/R_0$ after each cycle, and a path-dependent metric ($H$), which is defined as the area between the heating–cooling curve at each cycle. $H$ is dependent on the maximum fractional change in resistance $(\Delta R/R_0)_{\text{max}}$ and the maximum temperature change $(\Delta T)_{\text{max}}$ during the heating phase of each cycle. Figure 3 shows the schematic of the parameters calculated after each cycle. The normalized hysteresis ($H_N$) is defined as:

$$H_N = \frac{H}{(\Delta T)_{\text{max}} (\Delta R/R_0)_{\text{max}}} \tag{3}$$

where $H_N$ is the normalized hysteresis, $H$ is the area under the hysteresis loop, $(\Delta T)_{max}$ represents the maximum change in temperature achieved in each cycle, and $(\Delta R/R_0)_{max}$ is the maximum fractional change in electrical resistance associated with the temperature change.

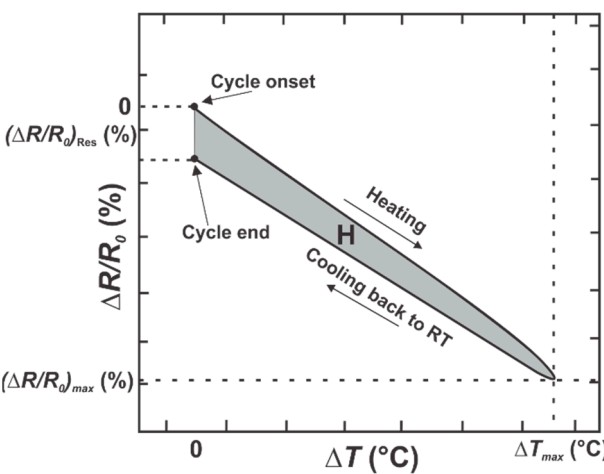

**Figure 3.** Schematic of the experimental parameters used for thermoresistive analysis.

*2.4. Swelling Characterization*

The network structure of the CNTY monofilament composites was studied by swelling experiments. The swelling of the cured ERHT and ERRT resins was done at ~25 °C using acetone (ChemPure Chemicals, 99.3%, Baltimore, MD, USA). In this work, the samples were approximately cut into 1 cm cuboids. The samples were dried at 80 °C and weighed ($W_0$). The dried samples were immersed in 20 mL of acetone for 24 h. The swollen samples were taken out and weighed ($W_{sw}$). The degree of swelling is defined according to (4) [31–33] and was calculated for the specimens.

$$d_{SW} = \frac{W_{SW} \ W_0}{W_0} \times 100\% \tag{4}$$

where $d_{sw}$ is the soluble fraction (%), $W_0$ is the initial weight of the sample, and $W_{sw}$ is the weight of the swollen sample.

*2.5. Scanning Electron Microscopy*

In order to investigate resin infiltration and interface morphology in the CNTY/ERHT and CNTY/ERRT monofilament composites, the fracture surfaces of CNTY monofilament composites were examined by scanning electron microscopy (SEM). SEM was conducted using a FEG-SEM field-emission MIRA3 TESCAN SEM (Kohoutovice, Czech Republic) operated at the acceleration voltage (ACC V) of 15 kV. The fracture surfaces of the specimens broken using the freeze-fracture method by means of the liquid nitrogen were coated with a thin layer of gold using the Quorum Q150R ES–magnetron sputtering machine (Laughton, East Sussex, UK).

**3. Results**

*3.1. Cyclic Thermoresistive Response of CNTY/Epoxy Monofilament Composites*

Figure 4 presents the cyclic thermoresistive response of CNTY/ERHT and CNTY/ERRT monofilament composites, respectively. Each specimen was subjected to heating above room temperature (RT~25 °C), which was ramped up at 0.8 °C/min until 80 °C (40 min), and cooling back to RT at 0.2 °C/min (4 h) for four continuous cycles. The first cycle was not considered for data acquisition in order to disregard the effect of moisture and residual thermal history on the electrical response of the CNTY monofilament composites [34]. The heating temperature coefficient of resistance ($\beta_H$) was calculated at heating zones for

both specimens according to Equation (1). Both specimens showed negative $\beta_H$ values of $(-7.07 \pm 0.53) \times 10^{-4}\ °C^{-1}$ for CNTY/ERHT monofilament composite specimens and $(-5.93 \pm 0.74) \times 10^{-4}\ °C^{-1}$ for CNTY/ERRT monofilament composite specimens.

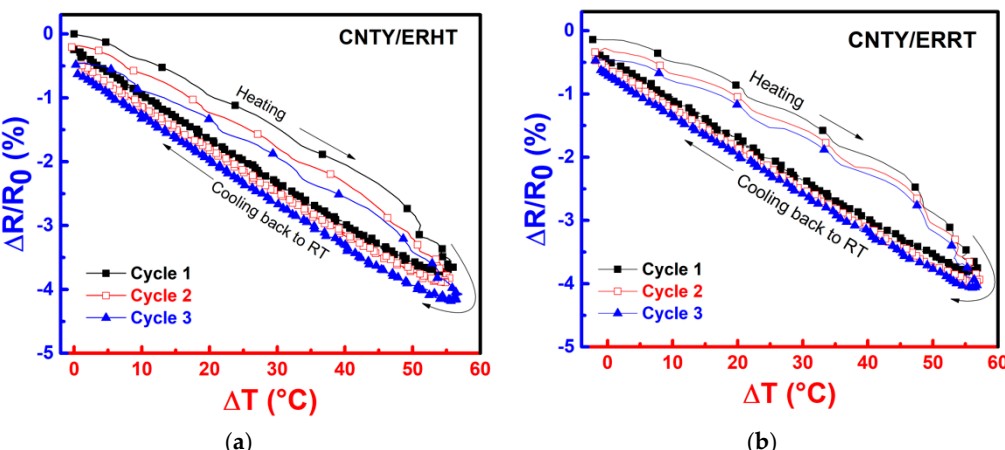

**(a)**          **(b)**

**Figure 4.** Cyclic thermoresistive response of CNTY/epoxy monofilament composites. (**a**) CNTY/ERHT, (**b**) CNTY/ERRT.

Table 1 shows the thermoresistive parameters obtained for CNTY monofilament composites during heating–cooling cycles. A higher temperature coefficient of resistance in absolute terms was observed for CNTY/ERHT in comparison to CNTY/ERRT.

**Table 1.** Summary of thermoresistive parameters obtained from heating–cooling cycles.

| Code | $\beta_H \times 10^{-4}$ ($°C^{-1}$) | $\beta_C \times 10^{-4}$ ($°C^{-1}$) | $(\Delta R/R_0)_{max}$ (%) | $(\Delta R/R_0)_{Res}$ (%) | $H_N$ (%) |
|---|---|---|---|---|---|
| CNTY/ERHT | $-7.07 \pm 0.53$ | $-6.63 \pm 0.24$ | $-4.27 \pm 0.55$ | $-0.65 \pm 0.40$ | $19.26 \pm 0.17$ |
| CNTY/ERRT | $-5.93 \pm 0.74$ | $-5.91 \pm 0.51$ | $-3.83 \pm 0.28$ | $-0.77 \pm 0.26$ | $22.63 \pm 0.32$ |

This negative value of $\beta_H$ is in agreement with the $\beta_H$ values reported for isolated CNTY [8,30,35] and carbon fibers [36,37]. Rodriguez–Uicab et al. [30] and Aliev et al. [8] observed a negative value of $\beta_H$ ($-8.4 \times 10^{-4}\ °C^{-1}$ and $-6.8 \times 10^{-4}\ K^{-1}$, respectively). The thermoresistive sensitivity was evaluated at cooling zones according to Equation (2). A negative and linear thermoresistive response was also observed for both composite materials, $-6.63 \times 10^{-4}$ ($\pm 2.41 \times 10^{-5}$) $°C^{-1}$ for CNTY/ERHT specimens and $-5.91 \times 10^{-4}$ ($\pm 5.10 \times 10^{-5}$) $°C^{-1}$ for CNTY/ERRT specimens. The maximum fractional change in resistance was slightly higher for CNTY/ERHT ($-4.27 \pm 0.55\%$). Both composite materials showed relatively small values of $(\Delta R/R_0)_{Res}$, yielding $-0.65\%$ for CNTY/ERHT and $-0.77\%$ for CNTY/ERRT specimens. These results correlated well with previous studies by Balam et al. [38] of an individual CNTY embedded in vinyl ester resin. In their results, the changes in the electrical resistance were attributed to the mobility restriction of the CNTY bundles [38], as well as to the mobility of electric charge carriers, leading to a quasilinear drop in electrical resistance of individual carbon nanotubes, resembling a semi-metallic behavior [39]. The hysteresis was calculated according to Equation (3) after each heating–cooling cycle. It was observed that the hysteresis values were higher for CNTY/ERRT specimens. This result is in agreement with that of Balam et al. [38]. Their results showed a hysteresis value of $21.6 \pm 3.4\%$ from RT to 100 °C. The negative thermoresistive behavior of CNTYs could be attributed to the intrinsic thermoresistivity of the CNTY and the contact resistance between CNTs and their bundles. The intrinsic thermoresistive sensitivity could be explained by quantum mechanics mechanisms, such as fluctuation-induced tunneling (FIT) and variable-range hopping (VRH), in which the electrical conductivity is an exponential function of temperature [40–43]. Increasing the temperature increases the density

of localized states, leading to an increase in mobility of electric charge carriers [40–43]. This behavior in CNTs resembles a combination of semi-metallic and small-gap semiconducting behavior that causes a drop in the electrical resistance of individual CNTs by increasing the temperature [43]. Another contributing factor in the thermoresistive response of the CNTY monofilament composites is the contact resistance between the CNT bundles affected by resin infiltration [13,30,31,38,44,45], leading to the restriction of the mobility of the CNT bundles due to the presence of epoxy resin within the CNT bundles [44,45]. The electron donor transfer from the resin to the CNTY may be another reason [44].

The higher temperature coefficient of resistance and lower hysteresis in CNTY/ERHT could be attributed to the higher crosslinking density in ERHT resin. The higher curing temperature in epoxy resin yields higher crosslinking density in the cured polymer [21,46,47].

Figure 5 shows the cyclic thermoresistive response of the CNTY monofilament composites under incremental temperature, from RT to 35 °C and cooling back to RT (Cycle 1), followed by heating from RT to 45 °C and cooling back to RT (Cycle 2), and, finally, from RT to 55 °C and cooling back to RT (Cycle 3). In order to disregard any thermal history, the first cycle was repeated and only the second cycle was used for data acquisition.

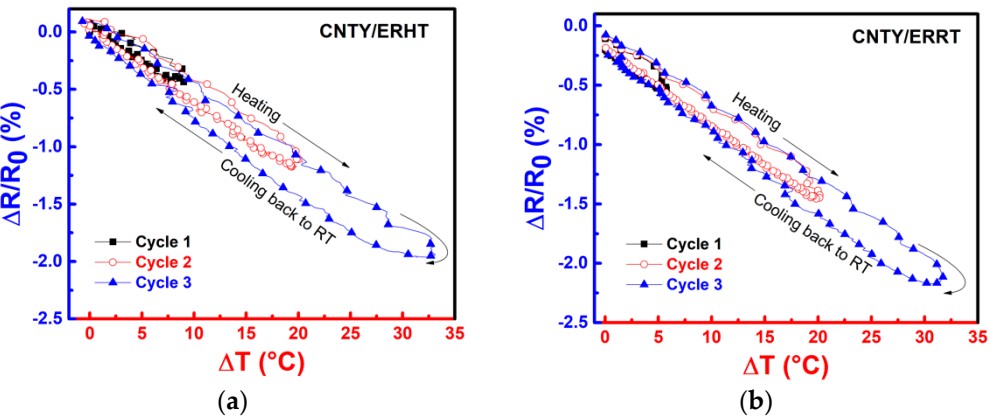

**Figure 5.** Cyclic thermoresistive response of CNTY/epoxy monofilament composites using incremental temperature. (**a**) CNTY/ERHT, (**b**) CNTY/ERRT.

Table 2 shows the cyclic thermoresistive parameters of CNTY/ERHT and CNTY/ERRT under incremental heating cycles. Slightly higher temperature coefficients of resistance in absolute terms were obtained for CNTY/ERHT in comparison to CNTY/ERRT, which is in agreement with the results in Table 1. $(\Delta R/R_0)_{max}$ increased by increasing $\Delta T_{max}$ in every cycle. The hysteresis was almost the same for both CNTY monofilament composites.

**Table 2.** Summary of thermoresistive parameters obtained from incremental heating cycles.

| Cycle | Material System | $\beta_H \times 10^{-4}$ (°C$^{-1}$) | $\beta_C \times 10^{-4}$ (°C$^{-1}$) | $(\Delta R^{(i)}/R_0^{(i)})_{max}$ (%) | $(\Delta R^{(i)}/R_0^{(i)})_{Res}$ (%) | $H_N$ (%) |
|---|---|---|---|---|---|---|
| 1 | | $-6.88 \pm 0.98$ | $-6.55 \pm 1.36$ | $-0.48 \pm 0.08$ | $0.08 \pm 0.01$ | $22.1 \pm 1.42$ |
| 2 | CNTY/ERHT | $-6.95 \pm 1.36$ | $-7.07 \pm 1.10$ | $-1.25 \pm 0.21$ | $0.1 \pm 0.02$ | $19.8 \pm 3.53$ |
| 3 | | $-6.72 \pm 0.87$ | $-7.0 \pm 1.14$ | $-2.02 \pm 0.13$ | $0.04 \pm 0.01$ | $20.5 \pm 8.49$ |
| 1 | | $-5.82 \pm 1.90$ | $-6.32 \pm 1.19$ | $-0.48 \pm 0.08$ | $-0.12 \pm 0.1$ | $22.3 \pm 1.75$ |
| 2 | CNTY/ERRT | $-6.17 \pm 0.25$ | $-6.19 \pm 0.22$ | $-1.40 \pm 0.02$ | $-0.12 \pm 0.08$ | $19.0 \pm 5.45$ |
| 3 | | $-5.97 \pm 0.51$ | $-5.97 \pm 0.01$ | $-2.07 \pm 0.04$ | $-0.17 \pm 0.1$ | $18.8 \pm 4.44$ |

### 3.2. Incremental Temperature Dwell Response of CNTY/Epoxy Monofilament Composites

Figure 6 shows the incremental heating dwell temperature–time program used for thermoresistive characterization of CNTY monofilament composites. The samples were heated from RT to 50 °C at 0.8 °C/min (cycle onset), and then the temperature was maintained at 50 °C for 3 h (D$_1$). The temperature was ramped up to 82 °C and maintained

at 82 °C for 3 h ($D_2$). Finally, the temperature was ramped up again from 82 °C to 110 °C and remained at 110 °C for 3 h ($D_3$). The maximum temperature was selected to avoid thermal degradation of the epoxy resin and analyze the post-curing effect in the thermoresistive response of CNTY. Specific temperatures of each dwell stage were selected to analyze two points distributed between RT and the maximum temperature. The specimens were cooled down to RT at 0.2 °C/min to complete the heating–cooling cycle (cycle end).

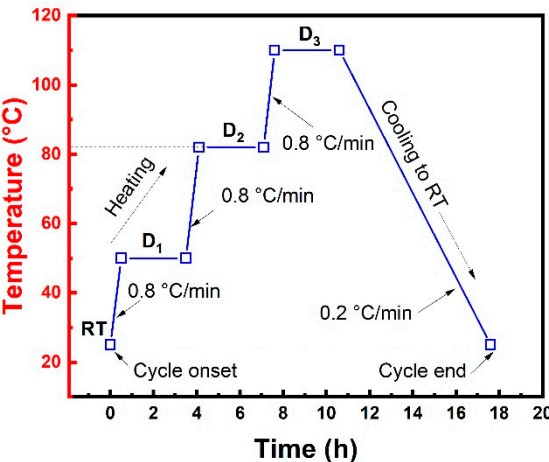

**Figure 6.** Temperature program for CNTY/epoxy monofilament composites.

Figure 7 shows the representative thermoresistive response curves of CNTY/ERHT and CNTY/ERRT monofilament composites, respectively. Both CNTY monofilament composites showed a negative thermoresistive response during the heating and cooling sections. It was observed that the fractional change in resistance did not return to the initial value by cooling down the specimens to RT. The $(\Delta R/R_0)_{Res}$ was −4.7% for CNTY/ERHT and −4.1% for CNTY/ERRT at the end of the cyclic experiment, in which $(\Delta R/R_0)_{Res}$ increased in comparison to that in the previous section. The decrease in $(\Delta R/R_0)_{Res}$ after the temperature dwells could be attributed to the effect of temperature dwells on the final network of the epoxy resin. Barton et al. [25] analyzed the effect of the initial curing conditions of epoxy resins. Their results showed that initial curing conditions determined the morphology of the early network formation and the post-curing conditions extended the degree of cure but also determined the distortion of the final network attributed to temperate changes. The temperature dwells and cooling back to RT with a lower rate of cooling could act as an isothermal annealing, leading to the stress relaxation and elimination of residual monomers [21,30,48–50].

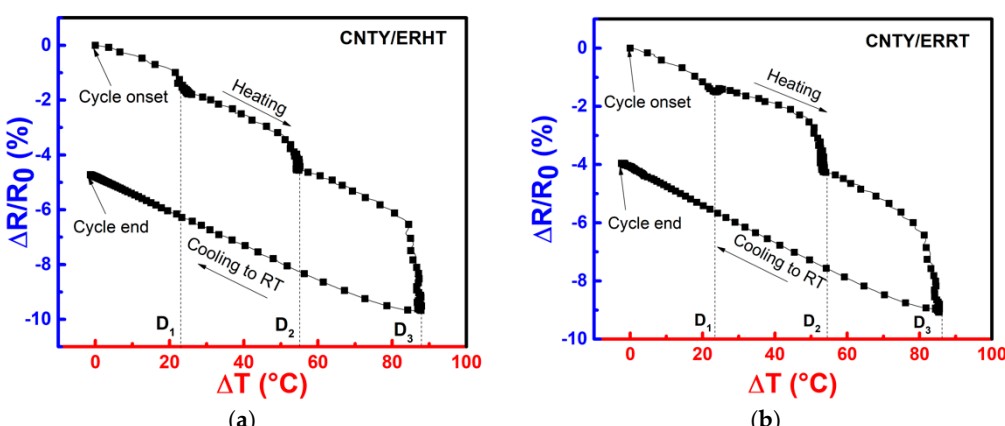

**Figure 7.** Thermoresistive response using incremental temperature dwells for CNTY/epoxy monofilament composites. (**a**) CNTY/ERHT, (**b**) CNTY/ERRT.

### 3.3. Swelling Behavior of Epoxy Resin Matrix

The swelling of the CNTY monofilament composites was conducted to compare the crosslinking density of the monofilament composites. Table 3 shows the degree of swelling calculated for the specimens. The solvent type has a significant role in the swelling behavior of the network. Acetone was used as the solvent due to the fact that the solubility parameter of the specimens is very close to that of acetone [20,50,51]. By comparing the degree of swelling calculated according to Equation (4), it was observed that both composites materials had a high degree of crosslinking due to a low solvent intake. The degree of swelling in ERRT was higher than that of ERHT, which could be due to the lower degree of crosslinking in ERRT. This value is agreement with previously reported results of the swelling of the epoxy resin [52]. The increase in crosslinking of the network reduces the movement of polymer chains and the free volume between the crosslinks, leading to a lower diffusion of the solvent [32,33,50]. It has been stated that curing of epoxy resin at elevated curing temperature in ERHT results in a denser three-dimensional network structure and decreases the free volume of the network, leading to a less efficient solvent uptake in ERHT [20,50,51].

**Table 3.** Degree of swelling of CNTY/ERHT and CNTY/ERRT.

| Solvent | $d_{sw}$ (%) | |
| --- | --- | --- |
| | **CNTY/ERHT** | **CNTY/ERRT** |
| Acetone | $0.41 \pm 0.09$ | $5.02 \pm 1.67$ |

### 3.4. Scanning Electron Microscopy

Figure 8 shows the representative SEM images of the fracture surfaces of CNTY/ER specimens at 4000× (image on the left) and 15,000× (image on the right). It is shown that the CNTY had a surface texture with high porosity, which may promote inter-bundle resin infiltration [44,45]. Figure 8a,b show the fractured surface morphology of a solid CNTY/ERHT specimen at 4000× and 15,000×, with the higher resolution focusing on the fiber/matrix interface, respectively. It was observed that the embedded CNTY was unbroken and there was no evidence of fiber/matrix debonding (flat fracture surface, with relatively small roughness). The fractured surface showed a homogeneous structure of the matrix. Figure 8b indicates a strong adhesion between the fiber and matrix. It is shown that the external layer of the embedded CNTY was densified and had a different structure in comparison to the center of the yarn. This could be attributed to the ingress of resin between the external bundles. The lower viscosity of ERHT at the curing temperature (130 °C), thermal and chemical shrinkage during the curing process, promotes the inter-bundle resin infiltration. Figure 8c shows the fracture surface morphology of a CNTY/ERRT specimen. It is shown that the CNTY is pulling out from the ERRT resin (sharp fracture surface, with relatively high roughness). The fiber/matrix debonding captured in Figure 8c indicates a weaker interface between the CNTY and resin. The weak interface (Figure 8d) could be attributed to the low evidence of resin infiltration due to more viscosity of the resin at the time of curing and curing at room temperature [30].

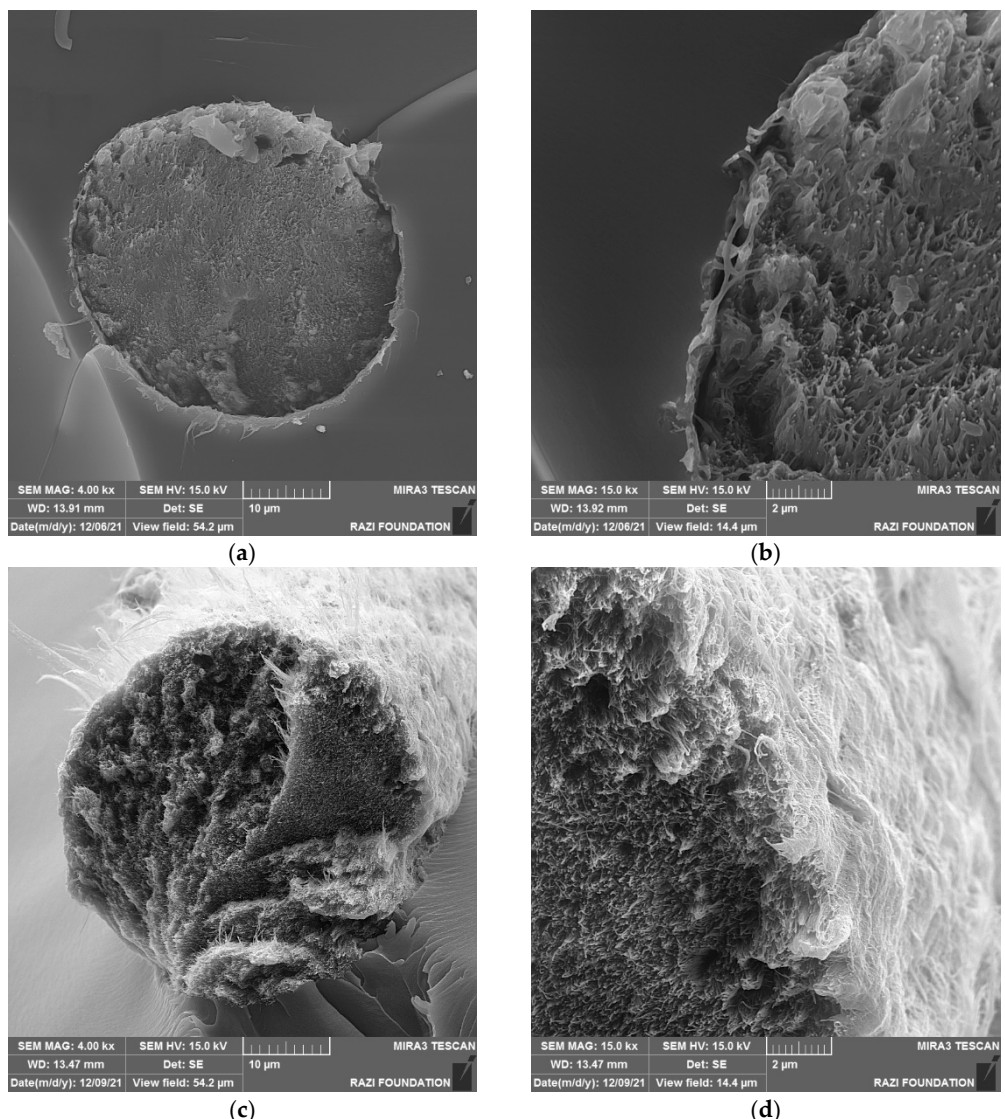

**Figure 8.** SEM images of CNTY/ERHT at (**a**) 4000×, (**b**) 15,000×. SEM images of CNTY/ERRT at: (**c**) 4000×, (**d**) 15,000×.

## 4. Conclusions

The thermoresistive response of carbon nanotube yarn (CNTY) monofilament composites was investigated by embedding the CNTY into two groups of epoxy resins with different curing mechanisms: CNTY/ERHT cured at 130 °C and CNTY/ERRT cured at room temperature (25 °C). Thermoresistive characterization of the CNTY monofilament composites was conducted under heating–cooling cycles above RT (25 °C to 100 °C). Both specimens showed a nearly linear response, with negative heating temperature coefficient of resistance values of $-7.07 \times 10^{-4}\,\text{C}^{-1}$ and $-5.93 \times 10^{-4}\,\text{C}^{-1}$. The thermoresistive sensitivity was slightly higher for specimens cured at a higher temperature (CNTY/ERHT). This negative thermoresistive behavior could be attributed to the increase in the mobility and density of electric charge carriers during heating, which leads to an exponential drop in the electrical resistance. The hysteresis loops calculated upon each heating–cooling cycle were lower in CNTY/ERHT (19.26%). Both composites materials showed a negative residual change in resistance of −0.65% and −0.77% for CNTY/ERHT and CNTY/ERRT, respectively. The potential wicking and resin infiltration within the CNT bundles, the change in the contact and tunneling resistance of individual carbon nanotube (CNT), interfacial and residual stresses, and volumetric and chemical shrinkage may have affected the ther-

moresistive response of the CNTY monofilament composites. The cyclic thermoresistive response of CNTY monofilament composites was also investigated at incremental heating–cooling cycles from RT to 35 °C (Cycle 1), RT to 45 °C (Cycle 2), and RT to 55 °C (Cycle 3) in order to investigate the effect of temperature on the sensitivity and hysteresis of the specimens. The temperature coefficients of resistance values were in agreement with previously reported values and higher in the CNTY/ERHT monofilament composites compared to the CNTY/ERRT ones. The hysteresis was almost the same for both monofilament composites. Finally, the effect of temperature dwells on the thermoresistive response of CNTY monofilament composites was studied using an incremental dwell temperature program. It was shown that the residual change in resistance decreased to $-4.7\%$ for CNTY/ERHT and $-4.1\%$ for CNTY/ERRT at the end of the increasing dwell temperature cycle. The temperature dwells and low rate of cooling could act as post-curing, leading to a change in the three-dimensional network of the composites. Several factors, including the intrinsic thermoresistivity of the CNTY, resin infiltration, change in the contact and tunneling resistance between the CNTs and bundles, and degree of crosslinking, influenced the behavior of the embedded CNTY in epoxy resins during the thermoresistive characterization.

**Author Contributions:** O.R.-U. conducted the majority of the work, provided the main idea, and analyzed the data. T.T. wrote the paper along with O.R.-U. and J.L.A. J.L.A. provided the laboratory facilities, expertise, technical input, and reviewed the final write-ups. All authors have read and agreed to the published version of the manuscript.

**Funding:** This research was supported by the National Aeronautics Space Administration (NASA) District of Columbia Space Grant Consortium (DCSGC) with grants NNX15AT64H S11 and NNX15AT64H S12 and 80NSSC20M0092 to Jandro L. Abot.

**Data Availability Statement:** The data have been obtained from the experimental results.

**Acknowledgments:** The authors would like to thank the Department of Mechanical Engineering at The Catholic University of America. Assistance of Behnam Rahmani (Razi Applied Science Foundation, Tehran, Iran) for the diligent SEM analysis is highly appreciated. The authors acknowledge the financial support from the National Aeronautics and Space Administration (NASA) District of Colombia Space Grant Consortium (DCSGC).

**Conflicts of Interest:** The authors declare no conflict of interest.

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
