# Peer review of "Effect of Curing Temperature of Epoxy Matrix on the Electrical Response of Carbon Nanotube Yarn Monofilament Composites"

_jcs, doi:10.3390/jcs6020043_

Round 1

Reviewer 1 Report

The reviewer thanks the authors for the submission and the Editor for the invitation. The reviewer feels that the paper is interesting, and it is within the scope of the Journal.

Specific comments:

  • In section 3, in point 4, The sem images show significant differences in the morphology of CNTY tubes of both CNTY/ERRT and CNTY/ERHT specimens. so, I am asking about the effect of temperature on the CNTY morphology?

Author Response

Reviewer 1

Authors sincerely thank the reviewer for the thorough reading of the manuscript and for making many constructive comments, which have contributed to its improvement. Authors have considered the reviewer’s comments, recommendations, observations, and requests in revising the manuscript. Authors’ response to reviewer’s comments and questions are presented here in italicized fonts.

Specific comments:

# 1a. In section 3, in point 4, The sem images show significant differences in the morphology of CNTY tubes of both CNTY/ERRT and CNTY/ERHT specimens. so, I am asking about the effect of temperature on the CNTY morphology?

R1a. The authors thank the reviewer for the comment. The CNTY fracture morphology of specimens cured at room temperature could be attributed to the yarn pullout during the specimens’ fracture as well as a weak interface, but not due to temperature effects. Two additional comments to this effect were added in section 3.4 of the revised manuscript.

Reviewer 2 Report

In general, the work is well structured and characterized. However, the work would be more complete if the authors compare the results of the CNTY/epoxy composites with the as-spun CNTYs without epoxies.

I include some more comments:

1) Authors should explain the reason why they choose that specific temperatures when design the heating-cooling and incremental heating-cooling cycles described in section 2.3.

2) The cooling back part of Figure 4 has higher number of experimental points than the heating part and authors should explain this difference.

3) What are the heating and cooling temperature coefficients of resistance of the as-spun CNTYs used in this study? Overall, the study would be more complete if the authors include the temperature-resistance analysis of as-spun CNTYs and compare it with the CNTYs epoxy composites.

4) Authors should explain the reason why they choose that specific temperatures when design the temperature program described in Figure 6.

Author Response

Reviewer 2

Authors sincerely thank the reviewer for the thorough reading of the manuscript and for making many constructive comments, which have contributed to its improvement. Authors have considered the reviewer’s comments, recommendations, observations, and requests in revising the manuscript. Authors’ response to reviewer’s comments and questions are presented here in italicized fonts.

In general, the work is well structured and characterized. However, the work would be more complete if the authors compare the results of the CNTY/epoxy composites with the as-spun CNTYs without epoxies.

I include some more comments:

#2a. Authors should explain the reason why they choose that specific temperatures when design the heating-cooling and incremental heating-cooling cycles described in section 2.3.

R2a.The authors thank the reviewer for the comment. The maximum temperature selected in section 2.3 was 80 °C that the thermoresistive response of the CNTY could not be affected by the thermal degradation of the epoxy resin. A description to this effect was added in section 2.3 of the revised manuscript.

#2b. The cooling back part of Figure 4 has higher number of experimental points than the heating part and authors should explain this difference.

R2b. The authors thank the reviewer for the comment. The heating and cooling rates used in section 3.1 (Figure 4) were 0.8 and 0.2 °C/min, respectively. Consequently, the heating time was about 40 min and the cooling time was about 4 hrs. These details were added in section 3.1 of the revised manuscript.

#2c. What are the heating and cooling temperature coefficients of resistance of the as-spun CNTYs used in this study? Overall, the study would be more complete if the authors include the temperature-resistance analysis of as-spun CNTYs and compare it with the CNTYs epoxy composites.

R2c. The authors thank the reviewer for the comment. The temperature coefficient of resistance of individual CNTYs has been extensively investigated by Rodriguez-Uicab et al. [30] and Balam et al. [38] and other authors. A brief description comparing these values to those in the current study and a mention to reference [30] was added in section 3.1 of the revised manuscript.

#2d. Authors should explain the reason why they choose that specific temperatures when design the temperature program described in Figure 6.

R2d. The authors thank the reviewer for the comment. The maximum temperature in Figure 6 was selected to avoid thermal degradation of the epoxy resin at high temperature (ERHT). The temperature difference between each dwell stage was about 30 °C, and the selected temperatures were 50 °C, 82 °C, and 110 °C, respectively. A description to this effect was added in Section 3.2 of the revised manuscript.
